# Research on Soil Nitrogen Balance Mechanism and Optimal Water and Nitrogen Management Model for Crop Rotation of Vegetables in Facilities

Xing Gan [1,2,*], Shiyu Sun [1,2], Haiyan Fan [1,3,4,*], Honglu Liu [1,3], Juan Zhang [1,3] and Zijun Ding [1,2]

1    Beijing Water Science and Technology Institute, Beijing 100048, China; ayussy@163.com (S.S.);
     liuhonglu@yeah.net (H.L.); zhangjuan0419@163.com (J.Z.); 221610010006@hhu.edu.cn (Z.D.)
2    College of Agricultural Science and Engineering, Hohai University, Nanjing 210098, China
3    Beijing Engineering Research Center for Unconventional Water Resources Development and Utilization and
     Water Conservation, Beijing 100048, China
4    College of Water Resources and Civil Engineering, China Agricultural University, Beijing 100048, China
*    Correspondence: 221610010009@hhu.edu.cn (X.G.); fanhaiyan0402@163.com (H.F.)

**Abstract:** Vegetable production is an important area of focus in China's agricultural structural adjustment plans, and it serves as one of the essential industries in the rural economy. Several studies have investigated how to optimize vegetable yield and quality through proper irrigation and fertilization to ensure efficient and sustainable development. The main objective of this paper is to examine the impact of different combinations of irrigation and nitrogen inputs on facility-grown vegetables under irrigation and fertilization conditions. Additionally, we aim to identify the optimal irrigation and fertilization regime that can enhance yield quality while also promoting environmental benefits. In this study, we focused on a white radish–tomato facility as the main research object. Using multiple regression and spatial analysis methods, we established three irrigation levels (W1: 100% ET0, W2: 85% ET0, W3: 70% ET0) and four nitrogen application levels (N0: no nitrogen, N1: high nitrogen, locally recommended nitrogen, N2: medium nitrogen, 85% N1, N3: low nitrogen, 70% ET0). We analyzed the effects of an irrigation nitrogen application on vegetable yield, nitrogen bias productivity, soil nitrogen surplus, and integrated N1 warming potential. Our experimental results showed that irrigation volume and nitrogen application had a considerable impact on the yield of facility-grown vegetables, and there was a positive correlation between irrigation water and fertilizer application and yield. By moderately reducing the irrigation volume and increasing nitrogen application, soil nitrogen surplus and nitrogen fertilizer bias productivity can be effectively improved. In addition, our study found that the integrated warming potential and the bias productivity of nitrogen fertilizer showed a quadratic relationship, which indicated that the integrated warming potential and nitrogen fertilizer bias productivity would first become larger and then decrease under the condition of increasing the irrigation volume and nitrogen application rate. By analyzing the difference between W2N2 and W1N1, we found that moderate water-saving and nitrogen reduction did not affect yield. Furthermore, it effectively improved the bias productivity of nitrogen fertilizer. Therefore, it is recommended that when the irrigation volume is between 560 and 650 mm and the nitrogen application rate is between 325 and 400 kg/hm$^2$ and more than 90% of the maximum value of yield, nitrogen fertilizer bias productivity can be achieved at the same time while also having a lower integrated warming potential. This range of irrigation and nitrogen application intervals is close to optimal. Our study provides a guiding basis for rotational soil nitrogen balance, optimal water, and nitrogen management of facility-grown vegetables. We propose an optimal water and nitrogen management strategy that is more efficient and sustainable under the plant culture model. This strategy provides a new way of thinking and methodology for high-quality production that is water-saving and fertilizer-saving while addressing the water and soil resource problems that exist in the current development of the vegetable industry.

**Keywords:** irrigation; nitrogen application; facility-grown vegetables; yield; nitrogen fertilizer bias productivity; regression analysis

## 1. Introduction

Vegetable production is an important part of China's agriculture. In 2021, China's vegetable sowing area was 21.986 million hectares, accounting for 13.03% of the total crop sowing area [1].

The cultivation of facility-grown vegetables in China has flourished in recent years, with the total area of production growing from 1.2 million hectares in 1999 to approximately 3.87 million [2,3] hectares today. However, the majority of these crops are shallow-rooted, meaning they are not particularly sensitive to soil moisture and nutrient levels. As a result, many farmers have become accustomed to using high levels of fertilizers and frequent irrigation to maintain their production processes. Unfortunately, this has led to excessive levels of nitrogen in the soil in many cases, with nitrogen fertilization often exceeding the actual demands of crops [4]. In some cases, it has been found that the average amount of nitrogen fertilizer used in vegetable fields in Beijing's suburban areas is as high as 1741 kg/hm$^2$ [5], which is much more than the demand for vegetables themselves [6]. Similarly, the amount of irrigation used is also excessive, with an annual average of 1307 mm [7]. This can lead to nitrogen residue in farmland and the accumulation of nitrate–nitrogen residue in facility fields, which can exacerbate problems with soil nitrogen leaching and increase the level of pollutants in the environment [8,9]. Furthermore, the excessive use of nitrogen fertilizers can result in the emission of greenhouse gases, such as $NH_3$ and $N_2O$, which contribute to global climate change [10].

The excessive use of nitrogen fertilizer can have a significant impact on the soil's carbon-to-nitrogen ratio. When the carbon-to-nitrogen ratio is low, it can lead to an increase in nitrogen loss, as well as the release of harmful substances, such as ammonia, which often generates unpleasant odors that can seriously affect the quality and effectiveness of fertilization [11]. Furthermore, excessive fertilization can also result in an overabundance of nitrogen infiltrating water sources in the form of nitrate or nitrous oxide, causing environmental pollution. Moreover, an excess of nitrogen in the soil can affect the nitrogen content and microbial population of the soil, ultimately resulting in the accumulation of nitrogen and ammonia, which hinders microbial growth. Nitrogen and ammonia are essential elements for microbial growth, and their high content in the soil can disrupt microbial balance, leading to negative effects on soil ecosystems [12,13].

To address these issues, it is vital to develop a water–fertilizer intercropping model that can achieve sustainable agricultural development and increase yields while also ensuring environmental safety. There have been numerous studies on the reduction in and control of nitrogen for facility-grown vegetables, with researchers proposing various fertilization methods to address the issue of soil nitrogen leaching [14–16]. Other studies have addressed issues, such as greenhouse gas emissions [17–19], soil nitrogen utilization, and nitrogen balance [20,21]. The researchers also focused on identifying the optimal combination of irrigation and fertilization for different indicators of facility-grown vegetables. This has typically involved developing water and fertilization regression equations with water and fertilizer application as independent variables, and individual indicators as dependent variables [22–25]. Utilizing multiple regression methods through statistical solutions of extreme values, such equations can be used to identify optimal water and fertilizer application combinations [26]. The findings of Dai Ming et al. [27] demonstrated that varying amounts of irrigation and nitrogen fertilizer had differing degrees of influence on the physiological characteristics and yield of greenhouse cucumber fruits. Yuan Liping [28] employed grey correlation theory to comprehensively evaluate different levels of water and nitrogen supply, revealing that the optimal combination was an irrigation volume of 2270.6 m$^3$/hm$^2$ with a specific amount of the nitrogen application. Wang Lei et al. [29] combined the photosynthetic physiological characteristics, seed yield, and organic fertilizer yield increase the effect of winter wheat and determined that the best results were obtained with a fertility water supply of 500 mm and an application of organic nitrogen fertilizer at a rate of 180 kg/hm$^2$ for improved leaf photosynthetic physiological characteristics. However, these evaluation methods only compare two or three indicators between different

treatments and select a relatively better treatment through manual screening. When there are too many evaluation indicators, the implementation difficulty of the above methods increases, and the water and fertilizer combinations obtained cannot usually take into account all the evaluation indicators, which cannot achieve the optimal values of all indicators simultaneously. For the comprehensive evaluation of multiple indicators, theoretically, the optimal water and fertilizer combination that considers each indicator can be obtained by jointly solving the multiple regression equations of each indicator. Moreover, due to the large amount of field experiments, using multiple regression and spatial analysis methods can find overlapping areas of acceptable ranges for specific indicators [30,31], thereby obtaining the best combination of water and nitrogen use. Thomas et al. [32] employed this spatial analysis method to comprehensively evaluate the agronomic, economic, and environmental benefits of drip-irrigated cauliflower, identifying the water and nitrogen ranges corresponding to the near-overlapping areas of acceptable indicators. The optimal range was found to be 100–120 mm for water and 300–400 kg/hm$^2$ for nitrogen.

As the water–nitrogen coupling index is seldom influenced by environmental factors, this study aims to conduct a comprehensive evaluation of various facility vegetable indicators using multiple regression analysis. This evaluation will be based on five key factors: high yield, water conservation, high quality, fertilizer conservation, and environmental protection. In addition, we will incorporate the spatial analysis method introduced by Thomas et al. to explore the synergistic effect between water and nitrogen. By establishing a synergistic management model for water and nitrogen in farmland crops, we aim to achieve coordination and unity among water and nitrogen resources, the environment, and yield. This research has significant scientific value in reducing greenhouse gas emissions and controlling farmland surface pollution caused by nitrogen.

## 2. Materials and Methods

### 2.1. Overview of the Pilot Area

The experiment was conducted in greenhouse No. 5 of Yongledian Experimental Base of Beijing Irrigation Experiment Center Station, which is located in Yongledian Town, Tongzhou District, Beijing (39°20′ N, 116°20′ E), at an altitude of 12 m above sea level, with a multi-year average rainfall of 565 mm, a multi-year average water surface evaporation of 1140 mm, a multi-year average temperature of 11.5 °C, a frost-free period of 185 d, and groundwater burial depth about 8 m; soil texture in the test area is mainly loamy with an average bulk mass of 1.49 g/cm$^3$, soil pH is 7.9, and field water holding rate is 29%.

### 2.2. Experimental Design

This trial uses a white radish–tomato rotation with a trial period of September 2021– June 2022. The planting is shown in Table 1.

**Table 1.** Planting status.

| Growing Crops | Species | Date of Sowing (Planting) | Harvest Date | Spacing between Plants and Rows | Irrigation Methods | Flooding Cycle |
|---|---|---|---|---|---|---|
| White radish | Jetmax 1410 | 24 September 2021 | 10 January 2022 | 0.45 m × 0.4 m | Drip irrigation | 5 days/time |
| Tomatoes | Strawberry 3 | 18 February 2022 | 23 June 2022 | 0.45 m × 0.4 m | Drip irrigation | 5 days/time |

Two variables were used in the trial: Nitrogen ($N$) and irrigation (W), with irrigation based on the evapotranspiration of the reference crop ($ET_0$) and $N$ applied at the recommended local rate. Three irrigation levels were set: W1 (100% $ET_0$), W2 (85% $ET_0$), W3 (70% $ET_0$); N1 (high N, locally recommended $N$ application), N2 (medium N, 85% N1), N3 (low N, 70% N1) each treatment was replicated three times for a total of nine treatments, and a control treatment (70% $ET_0$, no fertilizer) was set for tomato. The treatments were set up

as shown in Table 2. The plot size was 7.3 × 5 m, and each plot had four beds, each with two rows, 7.3 m long, 0.9 m wide, 0.4 m between rows, and 0.45 m between plants, with 20 plants planted in each row.

**Table 2.** Experimental treatment settings.

| Growing Crops | Processing Number | Irrigation Volume | Nitrogen Application Rate/(kg-hm$^{-2}$) | Test Treatment |
|---|---|---|---|---|
| White radish | W1N1 | 100% ET$_0$ | 240 | High water and high nitrogen |
| | W1N2 | 100% ET$_0$ | 204 | Nitrogen in high water |
| | W1N3 | 100% ET$_0$ | 168 | High water low nitrogen |
| | W2N1 | 85% ET$_0$ | 240 | High nitrogen in medium water |
| | W2N2 | 85% ET$_0$ | 204 | Nitrogen in water |
| | W2N3 | 85% ET$_0$ | 168 | Medium water low nitrogen |
| | W3N1 | 70% ET$_0$ | 240 | Low water high nitrogen |
| | W3N2 | 70% ET$_0$ | 204 | Nitrogen in low water |
| | W3N3 | 70% ET$_0$ | 168 | Low water and low nitrogen |
| Tomatoes | W1N1 | 100% ET$_0$ | 491 | High water and high nitrogen |
| | W1N2 | 100% ET$_0$ | 417 | Nitrogen in high water |
| | W1N3 | 100% ET$_0$ | 344 | High water low nitrogen |
| | W2N1 | 85% ET$_0$ | 491 | High nitrogen in medium water |
| | W2N2 | 85% ET$_0$ | 417 | Nitrogen in water |
| | W2N3 | 85% ET$_0$ | 344 | Medium water low nitrogen |
| | W3N1 | 70% ET$_0$ | 491 | Low water high nitrogen |
| | W3N2 | 70% ET$_0$ | 417 | Nitrogen in low water |
| | W3N3 | 70% ET$_0$ | 344 | Low water and low nitrogen |
| | CK | 70% ET$_0$ | No fertilizer | Low water no fertilizer treatment |

### 2.3. Observation Indicators and Methods

Irrigation water was controlled and measured at the head of each plot in the trial by installing a mechanical rotary-wing water meter; each plot was fitted with a Trime measuring tube, and soil moisture content was measured in layers using a Trime-IPH portable soil moisture monitor; meteorological data, such as sunshine hours, temperature, relative humidity and radiation were collected in the trial area by means of a small HOBO weather station; fruit yield was obtained by weighing on an electronic balance.

Soil and plant nitrogen measurements, soil nitrate, and ammonium nitrogen content were determined using the AA3 continuous flow analyzer; plant total nitrogen was determined using the KDY-9830 Kjeldahl nitrogen tester.

Measurement of greenhouse gas emissions were determined. $NH_3$ emissions by ammonia release from soil were collected using the phosphoglycerine sponge aeration method [33]. Two sponges soaked in phosphoglycerine solution (a mixture of 50 mL phosphoric acid and 40 mL propanetriol fixed to 1 L) were placed in a PVC rigid plastic tube, with the lower sponge at the height of the bottom of the tube and the upper sponge flush with the top of the rigid plastic tube; samples were taken daily at 8 am. Samples were taken once a day for the first three days after fertilization, and thereafter as appropriate. The amount of ammonium nitrogen absorbed in the sponges was determined by UV spectrophotometry. The technology is non-invasive, enabling samples to be reused or subjected to further analysis and processing. Measurements can be swiftly conducted, facilitating the seamless integration of the instrument into experimental protocols. The equipment is user-friendly, necessitating minimal training before usage. Data analysis ordinarily calls for negligible processing, thereby requiring minimal user orientation. Moreover, the apparatus is generally inexpensive to procure and operate, rendering it readily accessible for deployment in numerous scientific laboratories.

$N_2O$, $CO_2$,and $CH_4$ emissions were measured by the static box method, which was used to collect samples [34] every 7 d. The collection time was between 8:00 a.m. and 10:00 a.m. each day and 100 mL of box gas was collected at 0 min, 5 min, 10 min, and 15 min after the box was covered, respectively. Samples were taken continuously for one week after fertilization, once a day. Gas concentrations were measured using a meteorological chromatograph. Gas chromatography (GC) stands out from other analytical methods due to its diverse range of applications. GC is capable of analyzing gases, liquids, and solids with exceptional sensitivity, detecting trace substances and performing quantitative analysis at the milligram level with an injection volume of less than 1 mg. In addition, GC offers a rapid analysis time, completing an analysis within minutes to tens of minutes, while maintaining ease of operation. One of GC's greatest strengths is its high selectivity, enabling the separation of similar substances and multi-component mixtures.

### 2.4. Data Processing and Methods

(1)    The $NH_3$ emission calculation [33].

$$F_1 = 0.1 \times \frac{tVC}{A} \qquad (1)$$

where $F_1$ is the daily ammonia emission, $kg \cdot hm^{-2}$; $V$ is the solution volume in mL; and $C$ is the nitrate–nitrogen concentration from the calibration curve based on the absorbance value of the soil sample, $\mu g \cdot mL^{-1}$.

(2)    The calculation of greenhouse gas emission fluxes, such as $N_2O$ ($CO_2$, $CH_4$) [34].

$$F_2 = H \times \frac{M \times P \times T_0}{V_0 \times P_0 \times T} \times \frac{dc}{dt} \times 1000 \qquad (2)$$

where $F_2$ is the $N_2O$ ($CO_2$, $CH_4$) gas emission flux, $\mu g \cdot m^{-2} \cdot h^{-1}$; $H$ is the height of the static box, cm; $M$ is the molar mass of $N_2O$ ($CO_2$, $CH_4$), $g \cdot mol^{-1}$; $V_0$ is the standard state $N_2O$ ($CO_2$, $CH_4$) molar volume, L; $P_0$ and $T_0$ are the standard atmospheric pressure and temperature, Pa, °C, respectively; $P$ and $T$ are the actual atmospheric pressure and

temperature at the sampling point, Pa, °C, respectively; and $dc/dt$ is the slope of the change in $N_2O$ ($CO_2$, $CH_4$) content over time during the sampling period, $c$ in ppm; $t$ in h.

(3)  The integrated warming potential calculation.

$CH_4$ and $N_2O$ emissions were converted into the combined greenhouse effect (GWP, kg $CO_2$·hm$^{-2}$·a$^{-1}$) in $CO_2$ equivalent, taking into account that 1% of $NH_3$-N would be converted into $N_2O$-N [35], so $NH_3$-*N* was converted into $N_2O$-N before calculating the combined warming potential (GWP). According to the IPCC 2013 report, the combined warming potential is calculated using the following formula [36]:

$$GWP = CO_2 + CH_4 \times 28 + (N_2O + 0.01NH_3 \times 44/28) \times 265 \tag{3}$$

(4)  Nitrogen fertilizer bias productivity calculation.

$$WUE = Y/M \tag{4}$$

where $Y$ is yield (kg/hm$^2$); $M$ is the amount of nitrogen applied (kg/hm$^2$); and $WUE$ is the nitrogen fertilizer bias productivity (kg/kg).

(5)  The calculation of soil nitrogen balance.

Inputs to the nitrogen balance are initial soil $N$, applied fertilizer $N$, flooded $N$, and mineralized $N$. The main outputs of the nitrogen balance are crop uptake, soil trapped $N$, $NH_3$ emissions, $N_2O$ emissions, and apparent losses of $N$ [37].

$$N_{\text{Initial or interception}} = \frac{H \times \rho \times C_{\text{Soil N}}}{10} \tag{5}$$

$$N_{\text{leaching}} = N_{\text{Harvesting (40~120cm soil layer)}} - N_{\text{Initial (40~120cm soil layer)}} \tag{6}$$

$$N_{\text{Mineralization}} = N_{\text{Crop}} + N_{\text{Initial}} - N_{\text{Interception}} - N_{\text{Irrigation}} \tag{7}$$

$$N_{\text{Other}} = (N_{\text{Initial}} + N_{\text{Fertilizer}} + N_{\text{Irragation}} + N_{\text{Mineralization}}) - (N_{\text{Interception}} + N_{\text{Crop}} + N_{\text{NH}_3} + N_{\text{N}_2\text{O}}) \tag{8}$$

$$N_{\text{Surplus}} = N_{\text{Fertilizer}} + N_{\text{Irrigation}} + N_{\text{Mineralization}} - N_{\text{Crop}} \tag{9}$$

where $N_{\text{Initial}}$ is soil–nitrogen accumulation before sowing, kg·ha$^{-1}$; $N_{\text{Interception}}$ is nitrogen accumulation in the 0–120 cm soil layer at harvest, kg·ha$^{-1}$; $H$ is soil thickness, cm; $\rho$ is soil bulk weight, g·cm$^{-3}$; $C_{\text{Soil N}}$ is soil $NO_3^-$-$N$ and $NH_4^+$-$N$ concentration, mg·kg$^{-1}$; $N_{\text{gonorrhea}}$ is 40–120 cm $N_3^-$-$N$ amount in the soil layer, kg·ha$^{-1}$; $N_{\text{Crop}}$ is plant nitrogen uptake, kg·ha$^{-1}$; $N_{\text{Mineralization}}$ is nitrogen mineralization in the control treatment, kg·ha$^{-1}$; $N_{\text{Irragation}}$ is nitrogen brought in by irrigation, mg-L$^{-1}$; $N_{\text{Other}}$ loss is other nitrogen losses, kg·ha$^{-1}$; $N_{\text{Fertilizer}}$ is applied fertilizer $N$, kg·ha$^{-1}$; $N_{\text{NH3}}$ is the nitrogen lost from *NH* emissions, kg·ha$^{-1}$; $N_{\text{N2O}}$ is the nitrogen lost from $N_2O$ gas emissions, kg·ha$^{-1}$; $N_{\text{Surplus}}$ is the input nitrogen minus the nitrogen absorbed and used by the crop, kg·ha$^{-1}$; and $N_{\text{leaching}}$ is the amount of nitrogen leached from the soil that is released to the local environment through irrigation water.

(6)  Processing and analysis of the test monitoring data using Microsoft Excel 2010 and MATLABR2022a.

## 3. Results and Analysis

### 3.1. Characterization of Nitrate–Ammonium Nitrogen Distribution in the Soil Profile

#### 3.1.1. Characteristics of Changes in Soil Nitrate–Nitrogen

Soil nitrate–nitrogen content increased over time at different soil depths, with more pronounced growth in tomatoes, indicating an increasing accumulation of nitrogen in the soil during the growing season; soil nitrate–nitrogen content increased with fertilizer application, with a more pronounced increase after the second fertilizer application in

tomatoes. The average nitrate–nitrogen content of white radish was reduced by 3–6% at a 15% and a 30% N reduction.

At equivalent irrigation levels, as shown in Figure 1, a 15% reduction in nitrogen application resulted in a 23% reduction in the average nitrate–nitrogen content of tomatoes compared to conventional fertilization. In addition, with a 30% or complete reduction in nitrogen application, the average nitrate–nitrogen content was reduced by 42% and 72%. Similarly, by reducing water use by 15% or 30% while applying the same amount of nitrogen, the average nitrate–nitrogen content of white radish and tomatoes decreased by 0.2–3.4%, and even 16–26%.

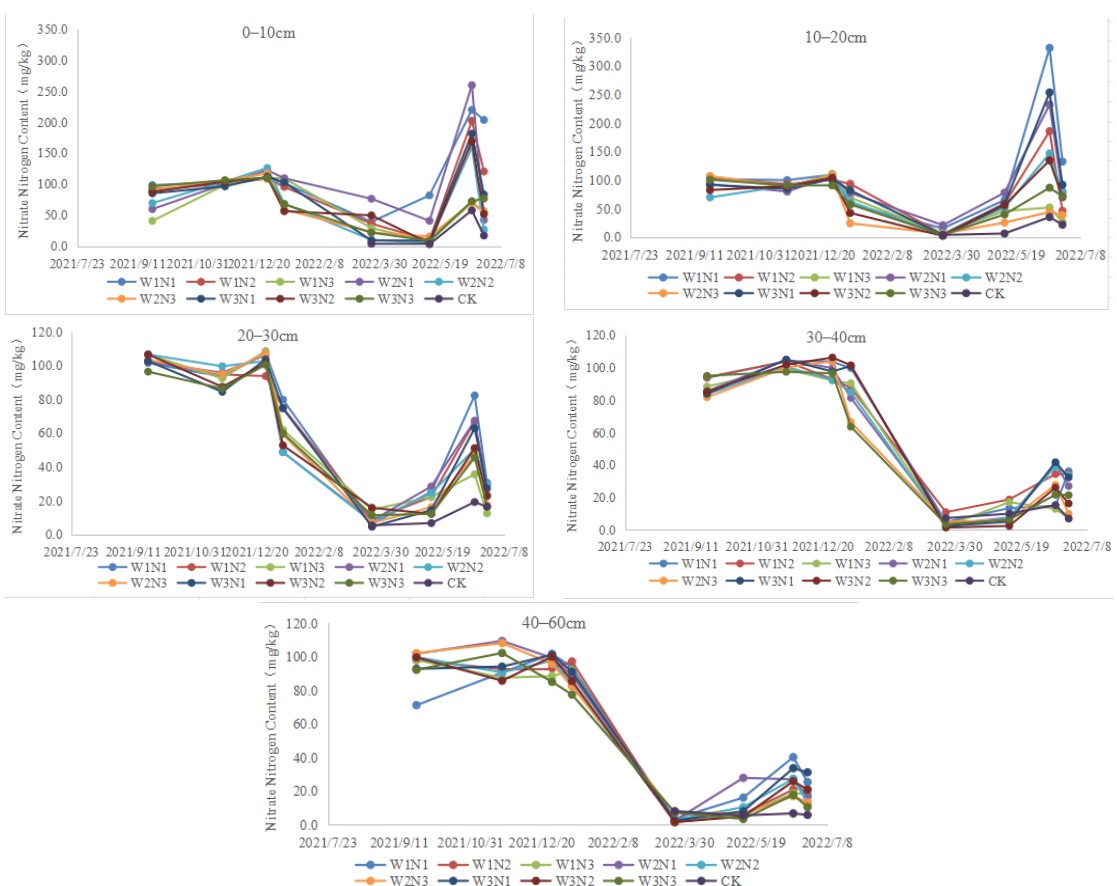

**Figure 1.** Variation in nitrate–nitrogen content of soils in different treatments.

### 3.1.2. Characteristics of Changes in Soil Ammonium Nitrogen

Soil ammonium nitrogen content fluctuated over time, with most treatments showing an increase in ammonium nitrogen content after fertilization, and a greater variation in soil ammonium nitrogen content was due to the application of basal fertilizer before tomato establishment. Figure 2 explains that at a 15% and a 30% N reduction, the average ammonium N content decreased by 9–30%; at the equivalent irrigation level, the mean concentration of ammonium nitrogen in tomatoes subjected to the unfertilized treatment declined by 15% in comparison to the low-nitrogen fertilizer, 32% in comparison to the moderate-nitrogen fertilizer, and 58% in comparison to the high-nitrogen fertilizer. Compared to normal irrigation levels, reductions of 8–16% in ammonium nitrogen content were observed for both white radish and tomato plants subjected to a 15% decrease in irrigation, while that same content decreased by 15–36% when irrigation was reduced by 30%.

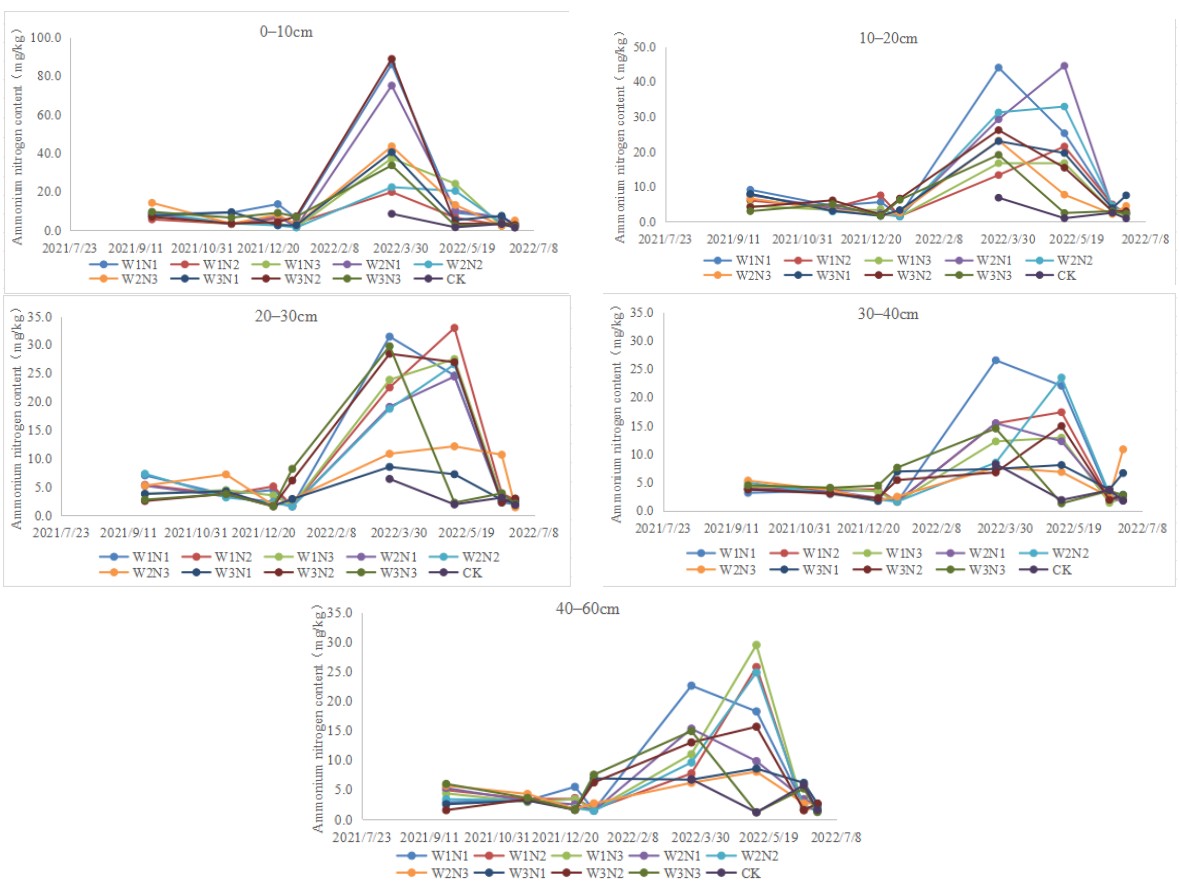

**Figure 2.** Changes in ammonium nitrogen content of soils in different treatments.

### 3.2. Plant Growth and Nitrogen Uptake in Different Water–Nitrogen Treatments

During the growing season, the root length and thickness of white radish increased continuously with time and tended to increase steadily at the end of the reproductive season. The nitrogen content of each organ of white radish during the whole growing season is shown in Figures 3 and 4. The nitrogen content of leaves tends to decrease continuously with the growth of the crop as a whole, with the average nitrogen content of leaves decreasing from 46.5 g/kg to 29.4 g/kg; the nitrogen content of fleshy roots tends to fluctuate and decrease with the growth of the crop as a whole, with the average nitrogen content of fleshy roots decreasing from 27.9 g/kg to 23.3 g/kg.

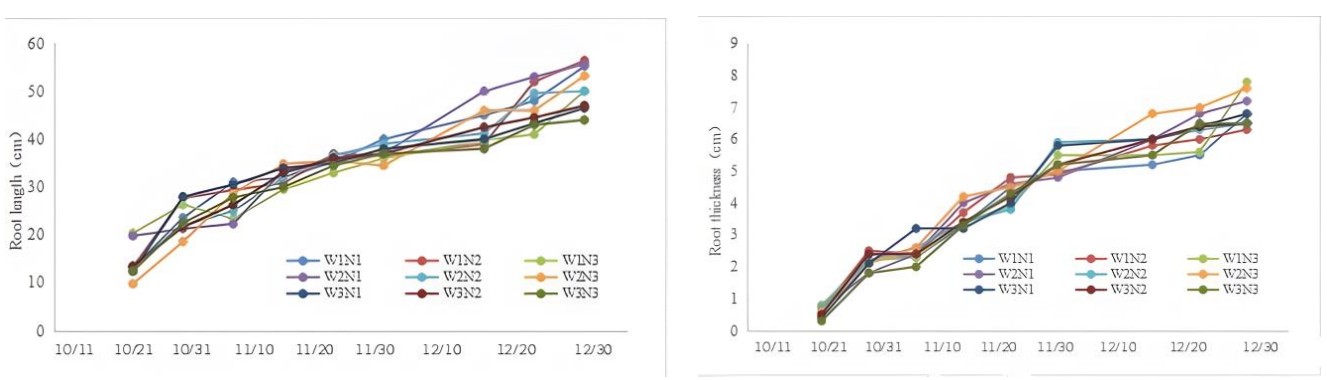

**Figure 3.** Changes in root length and thickness of white radish in different treatments.

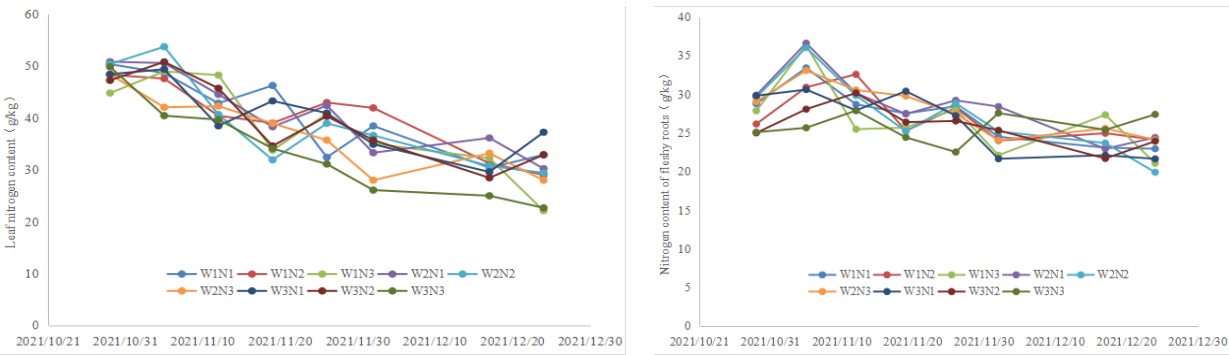

**Figure 4.** Changes in nitrogen content of various organs of white radish plants in different treatments.

As shown in Figure 5, during the growing season, the dry weight of roots, leaves, and stems of tomatoes increased with plant growth and development, with the most significant increase after the first fertilizer (22 March); the dry matter weight of fruits showed a trend of first increasing and then decreasing, reaching a peak around 25 May. The percentage of nitrogen in the roots of the plants showed a trend of decreasing and then increasing with the growth and development of the plants, with the most obvious increase after the first fertilizer application; the percentage of nitrogen in the leaves showed a trend of decreasing with the growth and development of the plants, while the percentage of nitrogen in the fruits kept increasing; the percentage of nitrogen in the stems was relatively stable throughout the whole reproductive period of the plants, with a certain increase after the three fertilizer applications.

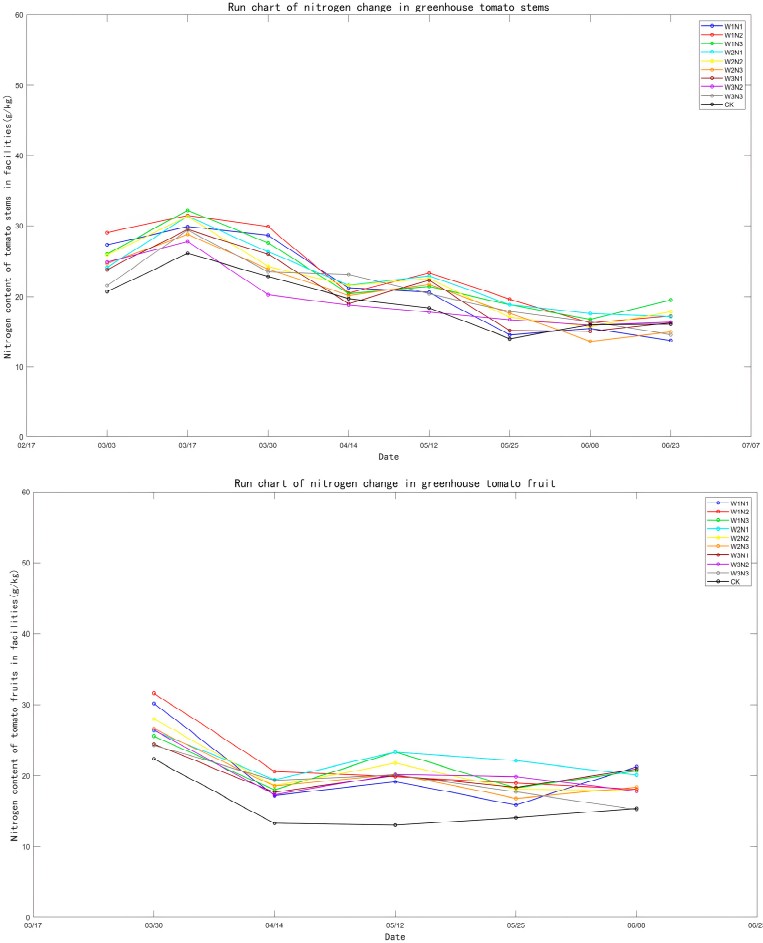

**Figure 5.** *Cont.*

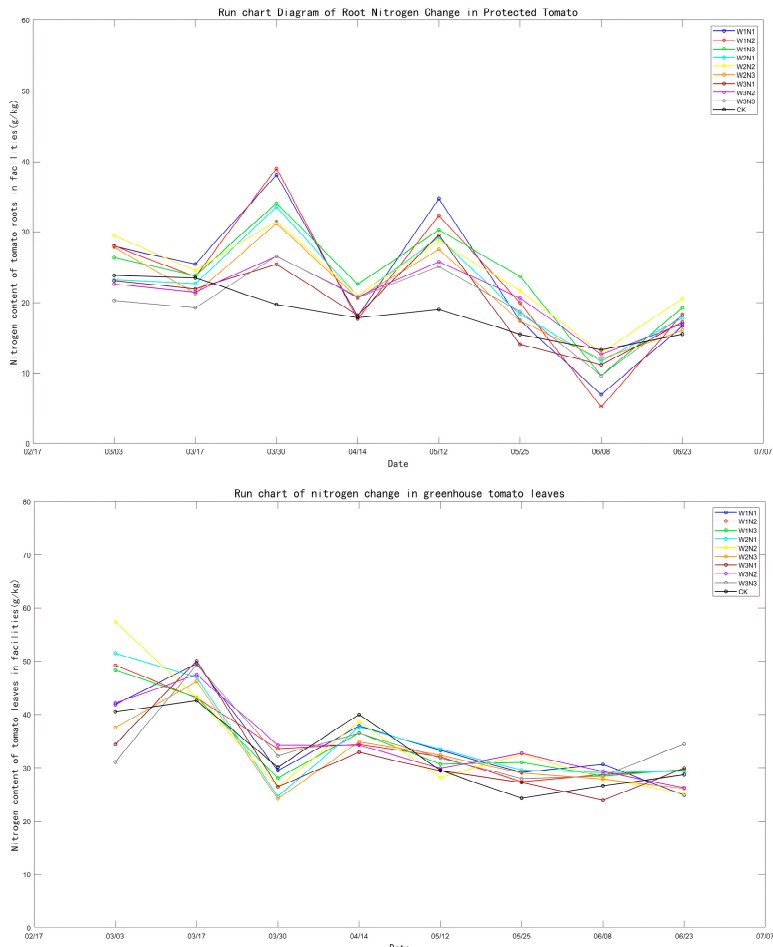

**Figure 5.** Changes in nitrogen content of various organs of tomato plants in the facility.

Figure 6 shows that in terms of the proportion of nitrogen in each part of the plant, the leaf nitrogen content was significantly higher than that of the other parts, with white radish showing overall leaf nitrogen > fleshy root nitrogen content and tomato showing overall leaf nitrogen > fruit nitrogen > stem nitrogen > root nitrogen content.

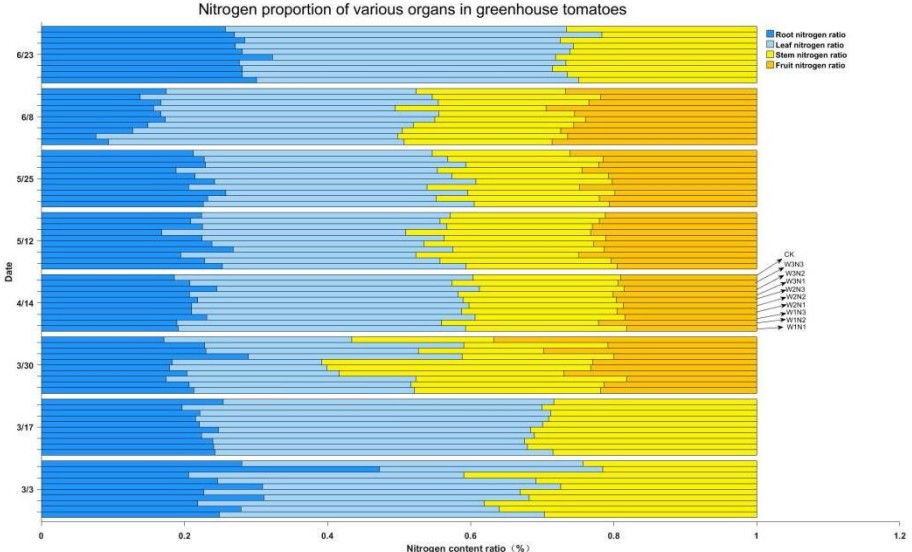

**Figure 6.** Percentage of nitrogen content in each organ of tomato plants at the facility.

### 3.3. Characterisation of Greenhouse Gas Emissions

3.3.1. Analysis of the Impact of $NH_3$

In Figures 7 and 8, the emission rate of $NH_3$ is significantly affected by fertilizer application, and the emission rate of $NH_3$ tends to rise first and then decrease after each fertilizer application, with the peak generally on the first to third day after fertilizer application. The cumulative emission of $NH_3$ gradually increases with crop growth; the cumulative emission of $NH_3$ is 4.8–6.0 kg/ha for white radish and 19.9–27.1 kg/ha for tomato during the whole fertility period.

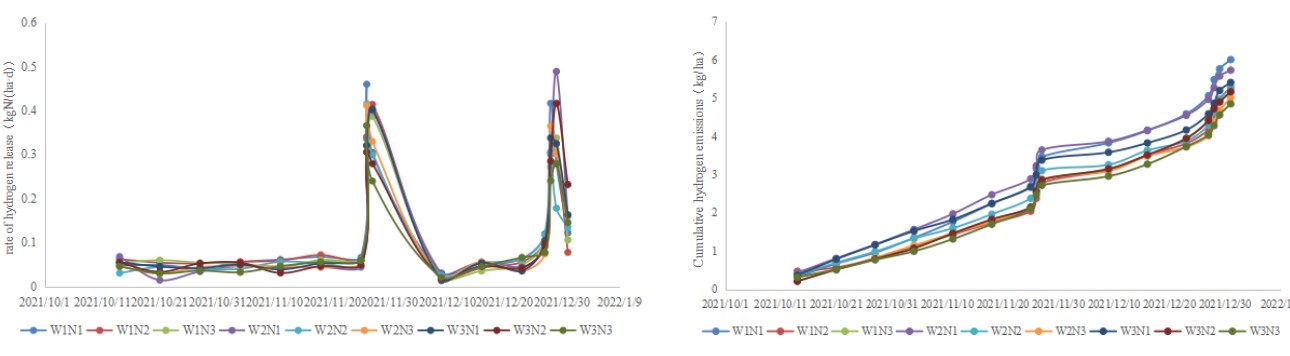

**Figure 7.** Emission rates and cumulative emissions of $NH_3$ from white radish soils in facilities.

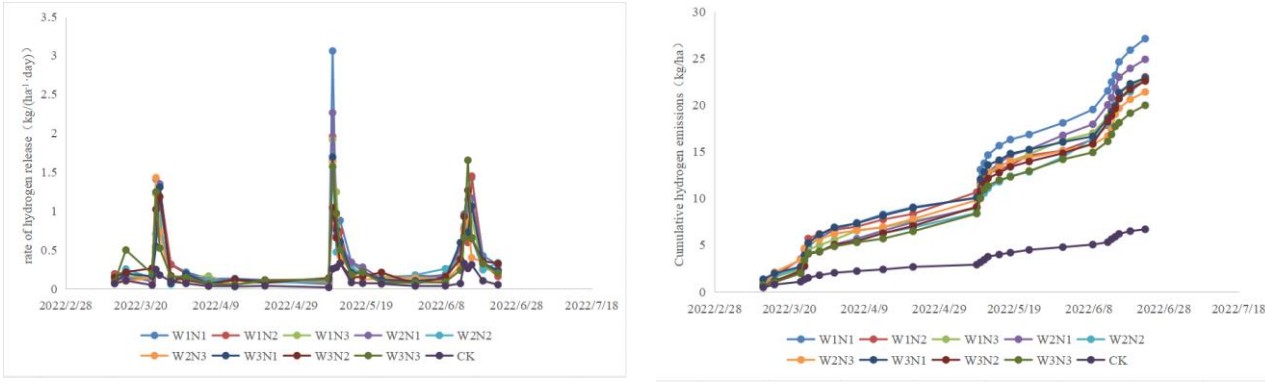

**Figure 8.** Rate and cumulative emissions of $NH_3$ from tomato soils in facilities.

The total amount of NH3 released from soils of white radish and tomato was reduced by 8–12% and 8–14%, respectively, when subjected to 15% and 30% reduction scenarios of a nitrogen application. Cumulative emissions of $NH_3$ from white radish and tomato soils were reduced by 3–6% and 5–10% under 15% and 30% water savings, respectively, and cumulative emissions of $NH_3$ from tomato soils were 3.0–3.4 times higher under the same irrigation conditions without fertilizer application.

3.3.2. Analysis of the Impact of $N_2O$

The emission rate of nitrous oxide ($N_2O$) was significantly impacted by various factors, including irrigation and fertilization. After irrigation and fertilization, the trend of $N_2O$ emissions showed an initial increase, followed by a decrease. Additionally, the cumulative emission of $N_2O$ was observed to increase gradually with crop growth. Treatment with water and nitrogen stimulated $N_2O$ emissions, and the peak of soil $N_2O$ emissions occurred after fertilization. The emission rate increased rapidly, with the peak reaching up to 60% of the cumulative emission of soil $N_2O$ during the entire incubation period. This can primarily be attributed to the hydrolysis of nitrogen fertilization, which provides a significant amount of an available nitrogen source for nitrifying and denitrifying microorganisms, thereby promoting soil $N_2O$ emissions. Moreover, the soil $N_2O$ emission rate reduced as the

soil moisture decreased, and the peak emission rate frequently appeared when the soil moisture condition was optimal. This could be explained by the fact that when both nitrogen and water sources are abundant, the soil water content increases, which in turn reduces the diffusivity of soil oxygen. This, in turn, facilitates denitrification by anaerobic microorganisms in the soil, leading to soil $N_2O$ emissions. This finding is consistent with the results of previous experiments conducted by Wang Y et al. [38]

Figures 9 and 10 explain that the total $N_2O$ emissions over the entire reproductive period were 1.2–1.8 kg/ha for white radish and 2.6–3.7 kg/ha for tomato. The implementation of a 15% and a 30% reduction in nitrogen resulted in a 17–24% and 6–20% reduction, respectively, in the cumulative soil $N_2O$ emissions for white radish and tomato. Similarly, a 5–11% and 7–9% reduction in the cumulative soil $N_2O$ emissions for white radish and tomato were observed when adopting 15% and 30% water savings. Moreover, the soil $N_2O$ emissions from fertilized treatments were 1.7–2.3 times higher than those from non-fertilized treatments under the same irrigation conditions.

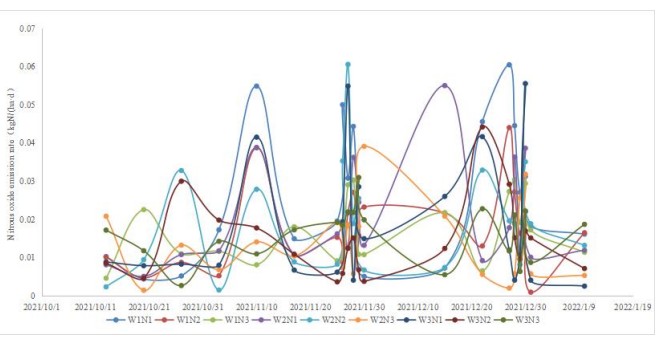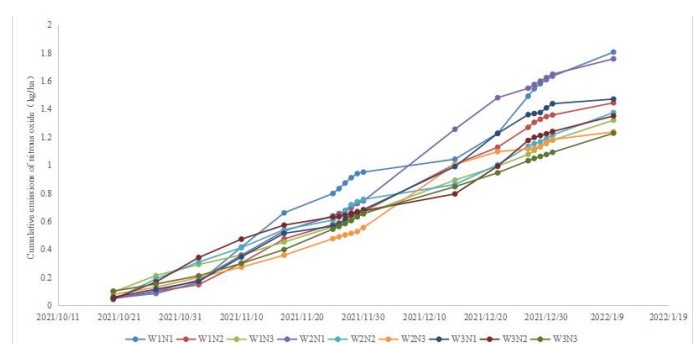

**Figure 9.** Rate and cumulative emissions of $N_2O$ from white radish soils in facilities.

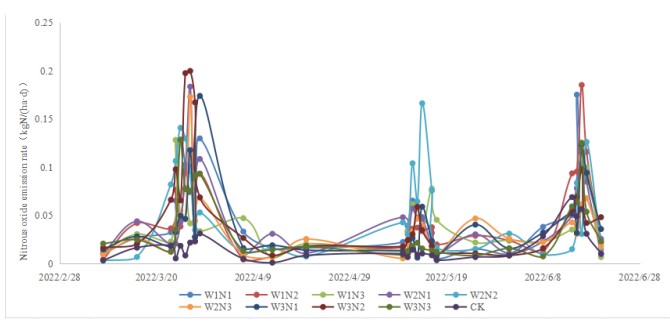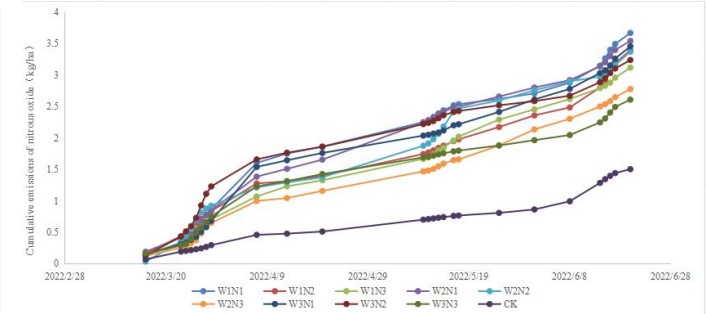

**Figure 10.** Rate and cumulative emissions of $N_2O$ from tomato soils in facilities.

### 3.3.3. Analysis of the Impact of $CO_2$

The emission rate of $CO_2$ is significantly influenced by irrigation and fertilizer application, with a trending increase followed by a decrease after an irrigation and a fertilizer application; the cumulative emission of $CO_2$ increases gradually with crop growth; the cumulative emission of $CO_2$ ranges from 3683 to 5151 kg/ha for white radish and 6536 to 15,186 kg/ha for tomato during the whole fertility period.

Figures 11 and 12 illustrate the contribution of white radish and tomato to $CO_2$ emissions. During the growth cycle of white radish, there was a 9% and 19% reduction in cumulative $CO_2$ emissions with a nitrogen reduction of 15% and 30%, respectively, while a 15% and a 30% water savings resulted in a 10% and 14% reduction in $CO_2$ emissions, respectively. By contrast, during the growth cycle of tomatoes, the cumulative $CO_2$ emissions were significantly influenced by the interplay between irrigation and fertilization. Specifically, the treatment with 100% irrigation and 85% nitrogen application had the maximum emissions, while the treatment with 70% irrigation and 100% nitrogen application had the minimum. Moreover, the fertilization treatment resulted in 1.5–3 times more $CO_2$

emissions than the treatment without fertilization but with the same irrigation conditions. Overall, it is apparent that the cumulative $CO_2$ emissions in the case of tomatoes were mainly influenced by the cooperative effect of irrigation and fertilization.

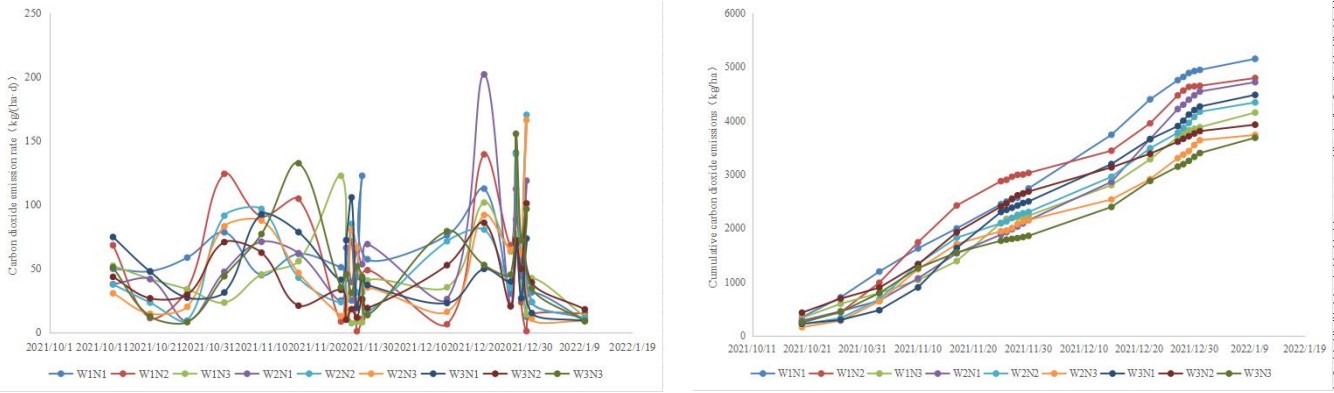

**Figure 11.** Emission rates and cumulative emissions of soil $CO_2$ from white radish in facilities.

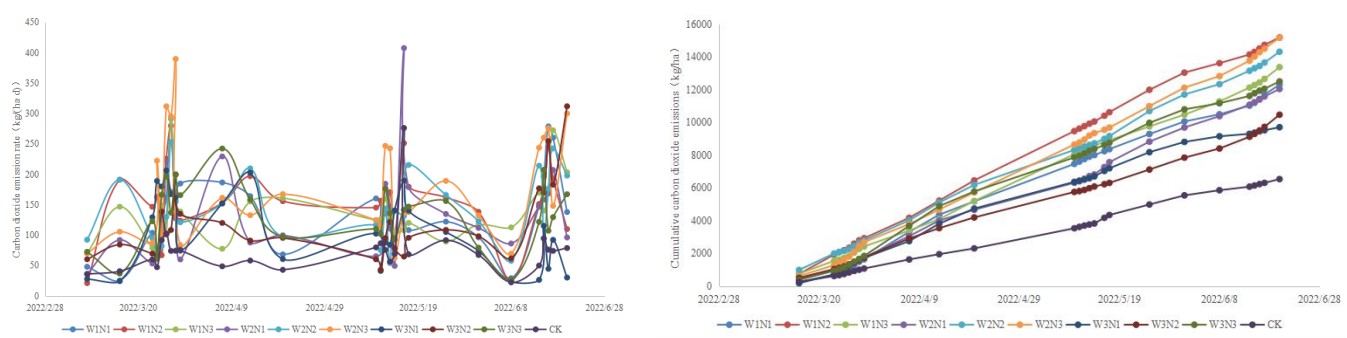

**Figure 12.** Facility tomato soil $CO_2$ emission rates and cumulative emissions.

### 3.3.4. Analysis of the Impact of $CH_4$

There was no significant difference in the trend of $CH_4$ emission rate among the different water–nitrogen treatments, and the emission rate of $CH_4$ varied slightly above and below the value of 0 during the whole reproductive period and did not respond significantly to external influences; the cumulative emission of $CH_4$ kept changing over time and remained at a stable level of break-even overall (Figures 13 and 14).

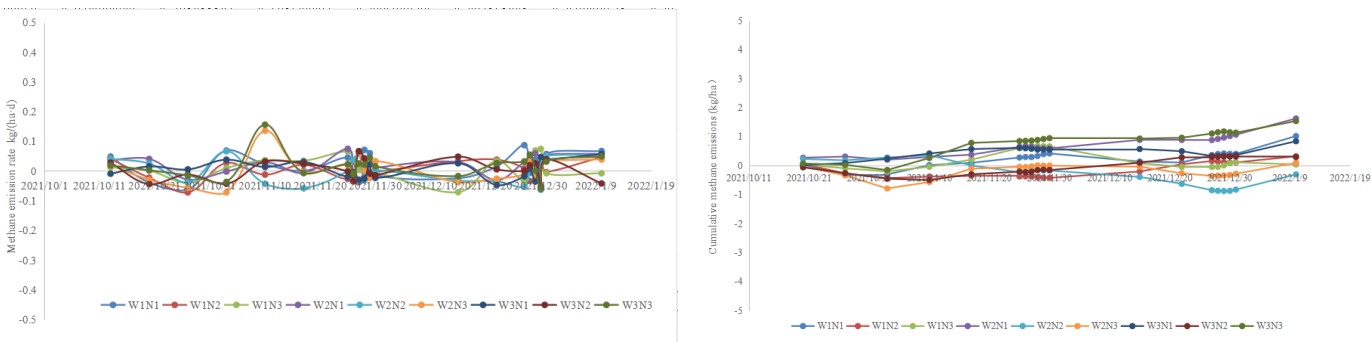

**Figure 13.** Emission rates and cumulative emissions of $CH_4$ from white radish soils in facilities.

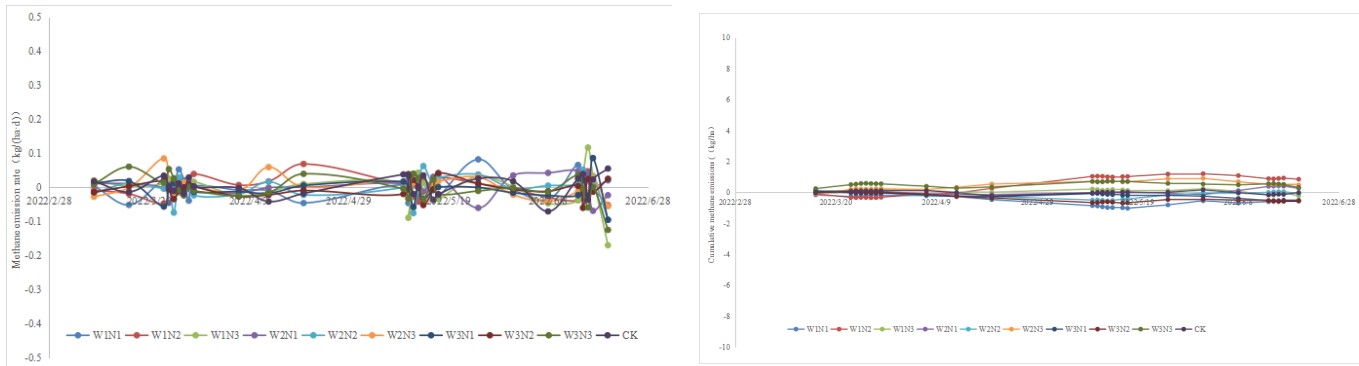

**Figure 14.** Rate and cumulative emissions of soil $CH_4$ from tomatoes in facilities.

### 3.3.5. Combined Warming Potential Effects

Converting $NH_3$, $N_2O$, and $CH_4$ emissions into a combined greenhouse effect with $CO_2$ as an equivalent, the warming potential of white radish was 3786–5477 kgCO$_2$·hm$^{-2}$ for the whole reproductive period, with $N_2O$ and $NH_3$ contributing 8.0–9.6% and $CH_4$ contributing only −0.2–1.1% based on Table 3.

The integrated warming potential of white radish was positively correlated with irrigation and nitrogen application levels during the growing season, both increasing with the amount of irrigation water and nitrogen application, and the effect of nitrogen application on the integrated warming potential was more significant. The combined warming potential was reduced by 7.9% and 14.1% with 15% and 30% water saving, respectively, and by 9.6% and 21.1% with a 15% and 30% nitrogen reduction, respectively.

**Table 3.** Calculation of the combined warming potential of white radish facilities.

| Processing | Combined Warming Potential GWP (kgCO$_2$·hm$^{-2}$) | | | | $N_2O$, $NH_3$ Contribution (%) | $CH_4$ Contribution Rate (%) |
| --- | --- | --- | --- | --- | --- | --- |
| | $N_2O$, $NH_3$ | $CO_2$ | $CH_4$ | Total | | |
| W1N1 | 503.28 | 4945.12 | 28.65 | 5477.04 | 9.19 | 0.52 |
| W1N2 | 404.51 | 4648.62 | 9.07 | 5062.20 | 7.99 | 0.18 |
| W1N3 | 371.17 | 3878.03 | 1.12 | 4250.33 | 8.73 | 0.03 |
| W2N1 | 489.44 | 4543.33 | 45.61 | 5078.38 | 9.64 | 0.90 |
| W2N2 | 386.10 | 4166.46 | −8.49 | 4544.06 | 8.50 | −0.19 |
| W2N3 | 348.50 | 3636.41 | 2.74 | 3987.65 | 8.74 | 0.07 |
| W3N1 | 412.04 | 4263.67 | 23.60 | 4699.31 | 8.77 | 0.50 |
| W3N2 | 379.22 | 3806.30 | 8.28 | 4193.79 | 9.04 | 0.20 |
| W3N3 | 345.40 | 3397.94 | 43.10 | 3786.44 | 9.12 | 1.14 |

Table 4 illustrates that the warming potential of tomatoes was 6962–16,205 kgCO$_2$·hm$^{-2}$ during the whole reproductive period, with the largest contribution from 100% ET$_0$ irrigation and 85% *N* application, 5.2–9.5% from $N_2O$ and $NH_3$, and only −0.01–0.15% from $CH_4$, accounting for a smaller contribution. The combined warming potential of the fertilized treatments was 1.5–1.9 times higher than that of the non-fertilized treatments under the same irrigation conditions.

**Table 4.** Calculation of the combined warming potential of tomatoes in facilities.

| Processing | Combined Warming Potential GWP (kgCO$_2 \cdot$hm$^{-2}$) | | | | N$_2$O, NH$_3$ Contribution (%) | CH$_4$ Contribution Rate (%) |
| --- | --- | --- | --- | --- | --- | --- |
| | N$_2$O, NH$_3$ | CO$_2$ | CH$_4$ | Total | | |
| W1N1 | 1084.37 | 12,298.42 | −15.26 | 13,367.53 | 8.11 | −0.11 |
| W1N2 | 994.72 | 15,186.00 | 23.83 | 16,204.55 | 6.14 | 0.15 |
| W1N3 | 920.85 | 13,368.66 | −6.44 | 14,283.07 | 6.45 | −0.05 |
| W2N1 | 1041.55 | 12,036.75 | 7.42 | 13,085.72 | 7.96 | 0.06 |
| W2N2 | 987.13 | 14,309.97 | −1.87 | 15,295.23 | 6.45 | −0.01 |
| W2N3 | 824.52 | 15,166.82 | 13.50 | 16,004.85 | 5.15 | 0.08 |
| W3N1 | 1010.57 | 9702.62 | −14.49 | 10,698.69 | 9.45 | −0.14 |
| W3N2 | 951.66 | 10,466.92 | −15.56 | 11,403.02 | 8.35 | −0.14 |
| W3N3 | 774.02 | 12,497.81 | 9.13 | 13,280.96 | 5.83 | 0.07 |
| CK | 426.04 | 6536.13 | −0.49 | 6961.68 | 6.12 | −0.01 |

*3.4. Analysis of Soil Nitrogen Use and Nitrogen Balance Characteristics*

3.4.1. Changes in Yield and Soil *N* Fertilizer Bias Productivity between Treatments

The results of the calculations of white radish yield and *N* fertilizer bias productivity for different treatments are shown in Table 5; under W1 and W3 irrigation levels, the yield of white radish decreased with increasing *N* application, with the largest treatment being W1N3, indicating that when the irrigation amount reached the crop growth demand, increasing the *N* application would inhibit the crop yield formation instead; under W2 irrigation level, the *N* application did not significantly affect the crop yield formation, and the yield of white radish was the largest in W2N3. The maximum yield of white radish was W2N3. Nitrogen fertilizer bias productivity increased with decreasing the *N* application, with *N* fertilizer bias productivity increasing by 29% and 62% at a 15% and 30% *N* reduction, and by 4% and 16% at a 15% and 30% water saving.

**Table 5.** Yield and *N* fertilizer bias productivity of white radish in facilities.

| Indicators | W1N1 | W1N2 | W1N3 | W2N1 | W2N2 | W2N3 | W3N1 | W3N2 | W3N3 |
| --- | --- | --- | --- | --- | --- | --- | --- | --- | --- |
| Capacity | 92,013.7 | 98,090.4 | 101,767.1 | 91,739.7 | 90,397.3 | 94,890.4 | 83,329.0 | 86,890.4 | 89,369.9 |
| Nitrogen fertilizer bias productivity | 383.4 | 480.8 | 605.8 | 390.1 | 443.1 | 564.8 | 347.2 | 425.9 | 532.0 |

The results of calculating the yield and *N* fertilizer bias productivity of different treatments of tomato in the facility are shown in Table 6. As can be seen from the table, tomato yield was greatest in the W2N1 treatment, which was 1.03–1.50 times that of the other treatments; tomato yield was significantly influenced by the amount of nitrogen applied, which increased with the increase in the amount of nitrogen applied. Nitrogen fertilizer bias productivity of tomato in different treatments decreased with increasing the nitrogen input and increased with increasing the water irrigation; nitrogen fertilizer bias productivity increased by 17% and 49% under a 15% and 30% nitrogen reduction; nitrogen fertilizer bias productivity increased by 3% and 4% under a 15% and 30% water saving.

3.4.2. Soil *N* Balance Analysis

As no control treatment was set for white radish, soil mineralized *N* could not be calculated, so this soil *N* balance analysis was carried out on tomatoes only. The largest *N* surplus was found in the 70% irrigation and 100% *N* application treatment, which was

188 kg/ha higher than the irrigated no-fertilizer treatment, and the proportion of *N* surplus to total *N* input in the different treatments ranged from 38% to 46%.

**Table 6.** Calculation of soil *N* balance for tomatoes in facilities.

| Nitrogen Balance Analysis (kg/ha) | | W1N1 | W1N2 | W1N3 | W2N1 | W2N2 | W2N3 | W3N1 | W3N2 | W3N3 | CK |
|---|---|---|---|---|---|---|---|---|---|---|---|
| Nitrogen input | Initial | 626.6 | 660.8 | 628.8 | 626.9 | 645.2 | 632.1 | 606.9 | 649.4 | 601.8 | 611.3 |
| | Irrigation *N* | 9.1 | 9.1 | 9.1 | 7.8 | 7.8 | 7.8 | 6.4 | 6.4 | 6.4 | 6.4 |
| | Fertilizer *N* | 491.0 | 417.0 | 344.0 | 491.0 | 417.0 | 344.0 | 491.0 | 417.0 | 344.0 | 0 |
| | Mineralized nitrogen | 657.8 | 657.8 | 657.8 | 657.8 | 657.8 | 657.8 | 657.8 | 657.8 | 657.8 | 657.8 |
| | Subtotal | 1784.5 | 1744.7 | 1639.8 | 1783.5 | 1727.7 | 1641.7 | 1762.1 | 1730.7 | 1610.0 | 1275.6 |
| Nitrogen output | Nitrogen uptake by the plant | 253.0 | 218.3 | 195.5 | 241.5 | 214.8 | 185.1 | 204.7 | 191.8 | 186.2 | 174.2 |
| | $NH_3$ Emissions | 27.1 | 23.0 | 22.8 | 24.9 | 22.7 | 21.4 | 22.9 | 22.5 | 19.9 | 6.7 |
| | $N_2O$ emissions | 3.7 | 3.4 | 3.1 | 3.5 | 3.4 | 2.8 | 3.5 | 3.2 | 2.6 | 1.5 |
| | Soil trapped nitrogen | 411.3 | 487.8 | 430.4 | 420.9 | 479.1 | 485.4 | 419.8 | 418.6 | 404.1 | 121.3 |
| | Other losses | 1089.4 | 1012.3 | 987.9 | 1092.7 | 1007.8 | 947.0 | 1111.4 | 1094.5 | 997.1 | 971.9 |
| | Subtotal | 1784.5 | 1744.7 | 1639.8 | 1783.5 | 1727.7 | 1641.7 | 1762.1 | 1730.7 | 1610.0 | 1275.6 |
| Soil *N* Surplus | | 904.9 | 865.7 | 815.5 | 915.1 | 867.8 | 824.5 | 950.6 | 889.5 | 822.1 | 490.1 |
| Soil nitrogen surplus/nitrogen export | | 0.507 | 0.496 | 0.497 | 0.513 | 0.502 | 0.502 | 0.539 | 0.514 | 0.511 | 0.384 |
| Capacity | | 135,461 | 127,749 | 120,650 | 139,361 | 132,264 | 125,170 | 129,652 | 126,557 | 124,822 | 92,768 |
| Nitrogen fertilizer bias productivity (kg/kg) | | 275.9 | 306.4 | 350.7 | 283.8 | 317.2 | 363.9 | 264.1 | 303.5 | 362.9 | 275.9 |

### 3.5. Relationship between Water and Nitrogen Inputs and Yield, Nitrogen Fertilizer Bias Productivity, Combined Warming Potential, and Soil Nitrogen Surplus

Multiple regression analysis was conducted using an irrigation and *N* application as independent variables and tomato yield, integrated warming potential, *N* fertilizer bias productivity, and soil *N* surplus as dependent variables. The results (Table 7) show that the irrigation and nitrogen application had significant effects on all dependent variables, with coefficients of determination $R^2 > 0.85$, indicating that the regression equations reflect the actual situation well.

**Table 7.** Regression equations for irrigation and *N* application versus tomato yield, *N* fertilizer bias productivity, combined heat gain potential, and soil *N* surplus.

| Dependent Variable | Regression Equation | $R^2$ | F | *p* |
|---|---|---|---|---|
| Capacity | $Y = -6537.5 + 385.593W - 28.9583N - 0.3726W^2 - 0.0908N^2 + 0.2716WN$ | 0.9942 | 137.9665 | 0.01 |
| Combined warming potential | $Y = -38,216 + 147.7152W + 14.8481N - 0.1193W^2 - 0.0802N^2 + 0.0563WN$ | 0.9485 | 14.7218 | 0.011 |
| Nitrogen fertilizer bias productivity | $Y = 77.8644 + 0.8034W + 0.4467N - 0.00080207W^2 - 0.0017N^2 + 0.00065927WN$ | 0.9452 | 13.8069 | 0.0124 |
| Soil *N* Surplus | $Y = 436.4663 - 0.1167W + 1.7565N + 0.00038858W^2 - 0.00029157N^2 - 0.0012WN$ | 0.9992 | 947.28 | 0.01 |

Note: W is the amount of irrigation water, mm; *N* is the amount of nitrogen applied, kg/hm$^2$.

The regression equations for the dependent variables in Table 7 were used to produce a plane projection (Figure 15) and to find the maximum values for each indicator and the corresponding amounts of irrigation and nitrogen applied (Table 8), from which it can be seen that the four indicators cannot be optimal at the same time (maximum yield and nitrogen bias productivity, minimum combined warming potential, and soil nitrogen surplus).

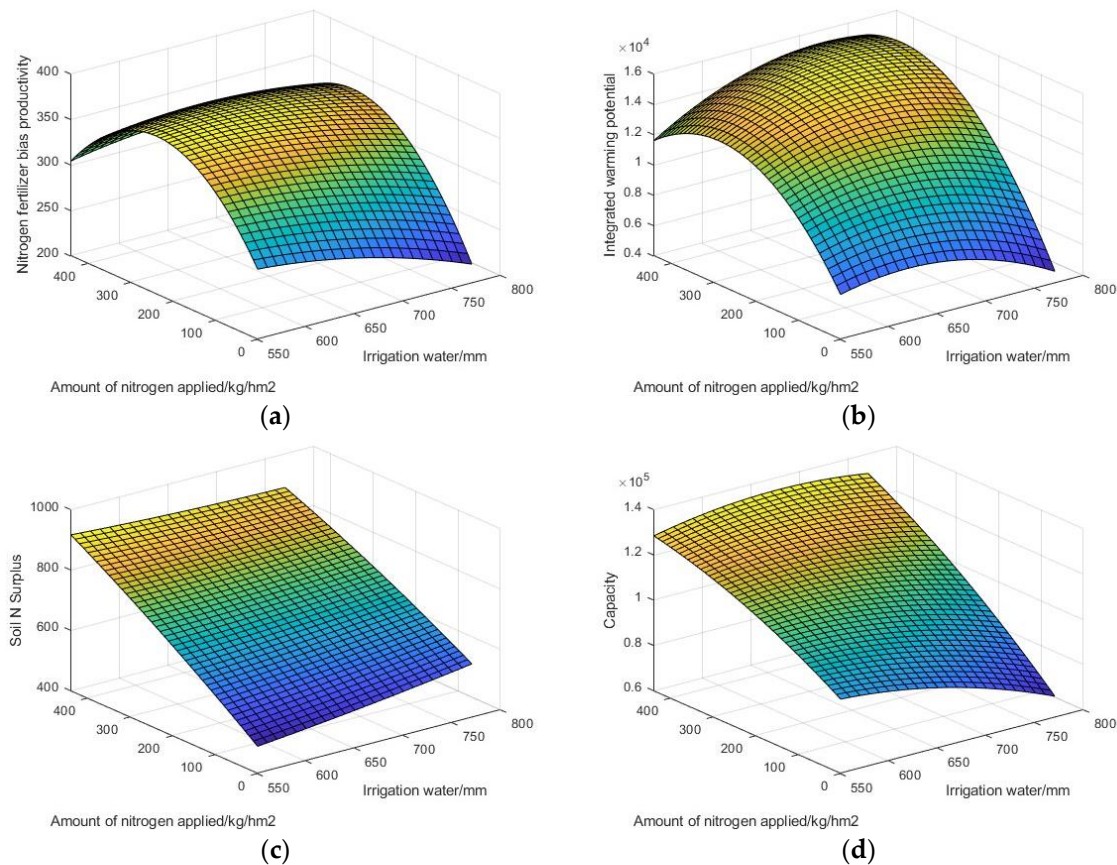

**Figure 15.** Relationship between irrigation and N application and yield, N fertilizer bias productivity, combined warming potential, and soil N surplus: (**a**) Soil N surplus; (**b**) nitrogen fertilizer bias productivity; (**c**) combined warming potential; and (**d**) yield.

The four indicators were analyzed according to Table 8 and Figure 15, and the four indicators were evaluated in the range of 5%, 10%, and 15% of the optimum values. It was found that yield and nitrogen fertilizer biased productivity overlapped in each range, and soil $N$ surplus and integrated warming potential deviated farther away and were not meaningful for finding water–$N$. Considering that the optimal values of soil $N$ surplus and integrated warming potential were taken as the minimum values, a contour projection of 90% to 95% of the optimal values of yield and $N$ fertilizer bias productivity indicators was adopted, and the optimal values of soil $N$ surplus and integrated warming potential indicators were expanded by 100% to 125%, and the wells were analyzed to find the intersection of their irrigation water and $N$ application.

**Table 8.** Maximum tomato yield, combined warming potential, $N$ fertilizer bias productivity, soil $N$ surplus and their required irrigation, and $N$ application rates.

| Dependent Variable | Value of Dependent Variable | Irrigation Volume W/mm | Nitrogen Application N/kg/hm$^{-2}$ |
|---|---|---|---|
| Yield max kg/hm$^{-2}$ | 138,050 | 669.3883 | 491 |
| Minimum integrated warming potential kgCO$_2$·hm$^{-2}$ | 6948 | 550.54 | 0 |
| Nitrogen fertilizer bias productivity max kg/kg$^{-1}$ | 375.4896 | 602.8673 | 248.28 |
| Minimum soil $N$ surplus kg/hm$^{-2}$ | 489.99 | 550.54 | 0 |

After a comprehensive analysis based on Figure 16, it was found that the soil *N* surplus indicator deviated significantly from the other three indicators after expanding by 100% to 125%, which was not meaningful for studying the relationship between water and fertilizer in tomato. Therefore, in the principle of maximizing tomato yield, an acceptable area of irrigation water and *N* application amount corresponding to 90% of the optimal value of yield and *N* fertilizer bias productivity indicators was considered, while the integrated warming potential was taken as a secondary factor, and the soil *N* surplus was not considered in this experiment. When the irrigation rate was 560–650 mm and the *N* application rate was 325–400 kg/hm$^2$, the yield and soil *N* surplus could reach more than 90% of the maximum value at the same time.

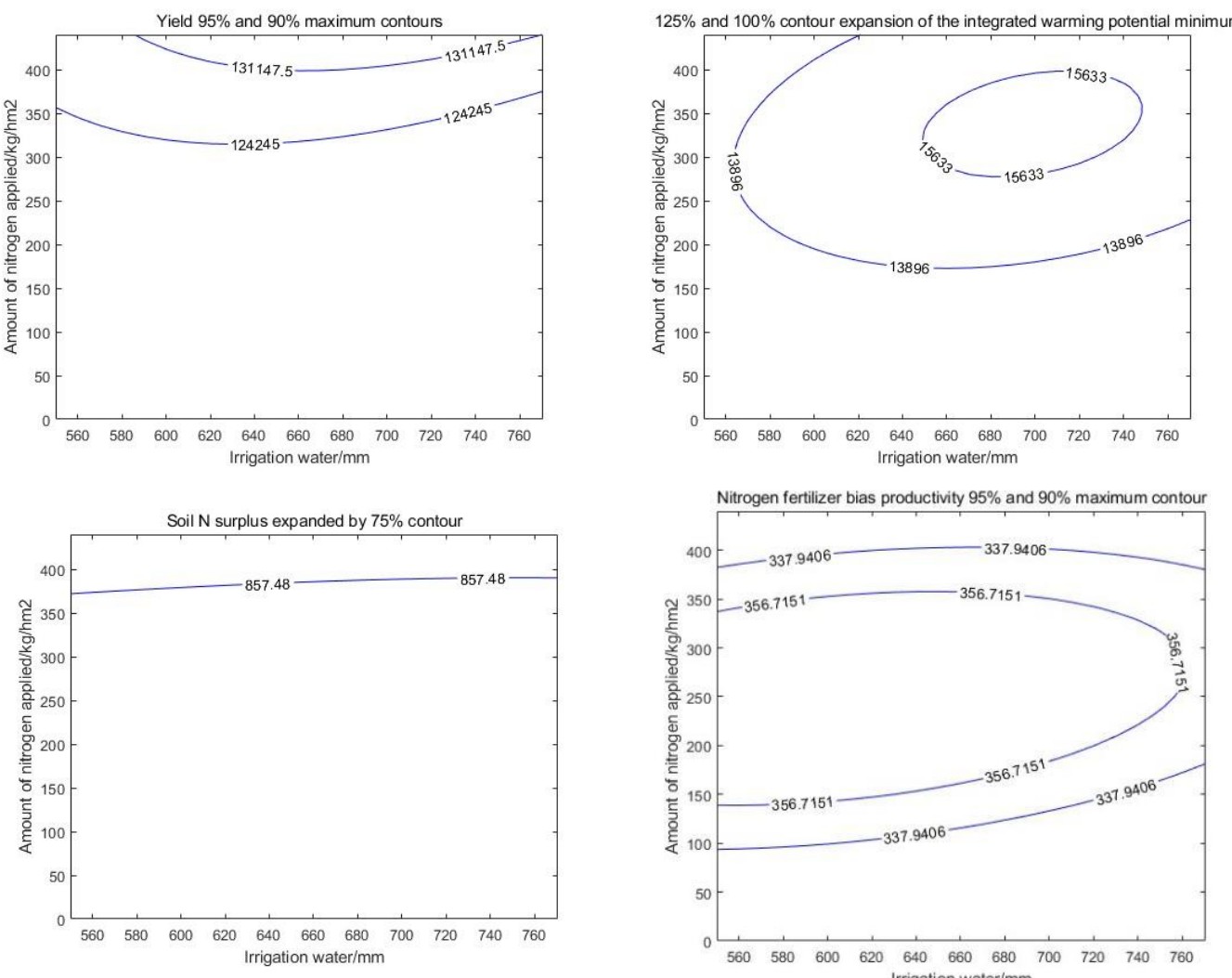

**Figure 16.** Relationship between water and fertilizer inputs and corresponding yields, soil *N* surplus, *N* bias productivity, and combined warming potential.

## 4. Discussion

### 4.1. Effect of Water and Nitrogen Availability on the Yield of Facility-Grown Vegetables

Research findings by Xu Guowei et al. [39] on the coupling of water and nitrogen for crops, such as rice, potatoes [40], and winter wheat [41] indicate that within a certain range of water and nitrogen levels, a crop yield increases with increased irrigation and nitrogen application. However, once the threshold is exceeded, yield tends to stabilize. In this study, the variation of vegetable yield was similar to the results of previous studies, but the yield increase showed a trend of increasing and then decreasing with the increase in water and nitrogen supply, which was consistent with the experimental findings of

Zhang Jingting [42]. It indicates that the appropriate amount of nitrogen application is favorable to increase crop yield and land contribution, probably because the appropriate amount of nitrogen fertilizer with the balance of each nutrient in the soil meets the nitrogen requirement for crop growth and helps to increase crop yield.

Under the N1 nitrogen application standard, the highest yield was achieved in the W2N1 test group, with an increase of 2.7% and 7.0% compared with the other two groups; under the W2 irrigation condition, W2N1 was still the highest yielding test group, with an increase of 4.9% and 10.2% compared with the other two groups, which indicates that the variation of yield in facility-grown vegetables is more sensitive to the irrigation amount. Under the same nitrogen application rate, the facility vegetable yield has a greater range of variation due to changes in the irrigation amount.

The productivity of facility-grown vegetables is directly linked to irrigation and fertilization. Positive correlation exists between the amount of irrigation and nitrogen application and the crop yield. This is because fertilization plays a significant role in promoting the development of the crop's root system, increasing water absorption, and improving the crop's photosynthetic rate. Adequate water conditions create an environment conducive to increasing stomatal conductance. Furthermore, the application of nitrogen under such conditions increases the content of chlorophyll in addition to promoting photosynthesis. This, in turn, enhances the crop's yield potential.

However, caution is required when supplying nitrogen to crops. Overapplication of nitrogen can cause the crop's yield to decline. This occurs for two main reasons. Firstly, the crop may fail to fully utilize the nitrogen, thereby leading to leaf roll-up. This reduces light energy utilization and increases the consumption of photosynthetic products leading to a reduction in yield. Secondly, excess nitrogen, which remains in the soil, affects the saturation of the soil with oxygen and gas exchange. This inhibits soil microbial activity, thus reducing the growth of soil microbial communities [43]. In turn, this is not conducive to the growth of crops and may lead to significant economic losses.

In summary, the growth of facility-grown vegetables is strongly associated with the adequacy of irrigation and fertilization. Adequate irrigation and nitrogen application levels are necessary to enhance plant growth, photosynthesis, and yield. It is important to balance these inputs and apply them in reasonable amounts to avoid negative impacts on yield. Hence, practitioners should consider the relevant factors to optimize their irrigation and fertilization strategy for facility-grown vegetable cultivars.

### 4.2. Effect of Water and Nitrogen Availability on Nitrogen Fertilizer Bias Productivity and Combined Warming Potential

The aim of agricultural water and nitrogen management is to maximize crop yields per unit of water and nitrogen used, and this is effectively achieved by developing effective water and nitrogen supply systems. Previous studies have shown that [44,45] *N* fertilizer bias productivity tends to increase and then decrease with increasing irrigation levels, while it tends to decrease with increasing *N* application levels. The present study showed that *N* fertilizer bias productivity tended to decrease with increasing irrigation levels and increase and then decrease with increasing *N* application levels. This may be due to the differences in water and nitrogen distribution caused by different irrigation methods, which in turn lead to differences in water uptake and utilization by the crop [46]. This may be due to the differences in water and nitrogen distribution caused by different irrigation methods, which in turn lead to differences in the uptake and use of water by the crop.

At the same time, increased irrigation and nitrogen application can increase the water content of the soil and the amount of substrate for nitrification and denitrification reactions, promoting microbial activity while increasing greenhouse gas emissions [47]. Lv Jindong [48] has shown that different greenhouse gas emissions are affected by changes in the water–nitrogen relationship to different degrees, mostly controlled by a single variable. In this study, the integrated warming potential tended to increase and then decrease with increasing *N* application and also tended to increase and then decrease with increasing

irrigation water, which is slightly different from previous studies. The reason for the difference in the results may be that irrigation and *N* application methods may have an impact on the combined warming potential; for example, subsurface drip irrigation, diffuse irrigation, and furrow irrigation have a greater impact on GHG emissions. The inconsistency of the findings is due to crop and regional differences [49]. The results are inconsistent due to crop and regional differences.

Similarly, it should be noted that the application of nitrogen fertilizer alone significantly reduces the diversity of soil microorganisms and alters their community structure [50,51]. This decrease in diversity is closely related to soil acidification and organic matter alteration due to ammonia fertilizer. As pointed out by Alvaro-Fuentes et al. [52] in their study, soil microbiomes are more susceptible to the level of nitrogen fertilizer application than to tillage patterns. Therefore, nitrogen fertilizer application is also responsible for the inconsistent results of experimental studies. It is important to note that soil nitrogen has a high bioefficacy, and the appropriate carbon-to-nitrogen ratio is crucial to the effectiveness of organic fertilizer application. The carbon-to-nitrogen ratio of the biomass in the soil indicates the ability of the soil to carry out nitrogen supply. The size of the ratio is inversely proportional to the bioefficacy of nitrogen, with higher bioefficacy observed at lower ratios [53]. Nitrogen fertilizer can be volatilized directly from the soil surface into the atmosphere after application. This causes an increase in the nitrogen content in the soil. Consequently, the carbon-to-nitrogen ratio decreases, limiting the source of energy required for microbial growth and reproduction. Slow fermentation temperature rises and excessive nitrogen converted into nitrogen oxides by soil microorganisms enter the atmosphere [54]. This results in large losses in organic nitrogen and accelerates the decomposition of organic matter. Additionally, it can emit unpleasant odors and cause harm to the environment. Furthermore, the release of methane from fields can also be harmful to the global atmosphere [55].

*4.3. Determination of the Optimum Water–Nitrogen Interval*

A comprehensive evaluation of yield and *N* fertilizer bias productivity concluded that yield and soil *N* surplus could simultaneously reach more than 90% of the maximum at 560–650 mm of irrigation water and 325–400 kg/hm$^2$ for tomatoes in the facility, and that the combined warming potential was small in this water–*N* zone. This irrigation and nitrogen application zone provides a basis for high yielding, high quality, and efficient water and nitrogen management for vegetables. Liu Junming et al. [56] also established the relationship between water–*N* inputs and yield, water–*N* use efficiency, and economic efficiency through a combination of multiple regression and spatial analysis.

Studies have shown that an excessive water and nitrogen supply does not necessarily lead to maximum values for all indicators but may have side effects. In optimizing water and nitrogen management systems, it is important to consider that for vegetables in different regions it is more important to take into account the conditions in the field to measure the range of water and nitrogen adaptation of vegetables.

## 5. Conclusions

In loamy soil environments, the balance between water and nitrogen significantly impacts the performance of facility-grown vegetables. Both an irrigation and a nitrogen application can affect vegetable yield, the productivity of nitrogen fertilizer, the integrated warming potential, and the surplus of nitrogen in the soil.

However, the crop yield increased with increasing the irrigation and nitrogen application, but showed a stable trend after exceeding the water–nitrogen threshold; nitrogen fertilizer bi-productivity showed a gradual decrease with increasing the irrigation level, and a trend of increasing and then decreasing with increasing the nitrogen application level; an irrigation and a nitrogen application would promote the growth of soil nitrogen surplus and increase the risk of nitrate–nitrogen leaching, taking into account the effects of tomato yield as well as the environmental effects and the use of irrigation. This coincides with

the suggestion by Bao Xuelian et al. [57] that nitrate–nitrogen accumulation is negatively correlated with irrigation water volume and positively correlated with fertilizer application.

A holistic evaluation of the tomato yield and the environmental impact indicates that an irrigation water amount ranging from 560 to 650 mm and a nitrogen application of 325~400 kg/hm$^2$ can ensure optimal yield. This range is also conducive to achieving high-yield, high-efficiency, and high-quality production of vegetables under the facility of a loamy soil environment. By enhancing water and nitrogen utilization, optimizing water and nitrogen management systems, and reducing environmental contamination, we can promote more sustainable and environmentally friendly production practices.

This study focused solely on establishing different treatments for irrigation water and nitrogen application levels. However, it did not delve deeper into the effects of irrigation water and the ratio of nitrogen, phosphorus, and potassium on crop growth. Future research should aim to quantify the proportion of irrigation and fertilizer application more precisely to provide a more scientifically sound and reasonable irrigation and fertilizer application system for facility-grown vegetables. It is critical to improve our understanding of these factors to optimize crop yield and quality.

**Author Contributions:** Methodology, S.S.; Resources, J.Z. and Z.D.; Data curation, H.L.; Writing—original draft, X.G.; Writing—review & editing, H.F. All authors have read and agreed to the published version of the manuscript.

**Funding:** This research was funded by [2021 Engineering Technology Centre Development Project] grant number [ERC-KF-2021-008-SZY].

**Data Availability Statement:** The authors confirm that the data supporting the findings of this study are available within the article. The data presented in this study are available on request from the corresponding author.

**Conflicts of Interest:** The authors declare no conflict of interest.

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
