# Peer review of "Research on Soil Nitrogen Balance Mechanism and Optimal Water and Nitrogen Management Model for Crop Rotation of Vegetables in Facilities"

_water, doi:10.3390/w15162878_

Round 1
Reviewer 1 Report
This manuscript by Gan Xing et al. presents a sophisticated model for the optimisation of water and nitrogen management in crop rotation under greenhouse conditions. It is underpinned by a wide range of quality experimental data. The paper is well written and generally interesting.
This is a well set up detailed experimental study with interesting and very relevant findings, aiming at establishing the balance between agricultural productivity and environmental impact, hence addressing one of the major challenges of modern-day agriculture.
The paper could further gain in impact if the authors made clear how generic their conclusions are and where findings are specific for the soil and climatological conditions of the tests.
I have some difficulties with following the presentation of the experimental results as discussed in section 3. Given the multi-parameter nature of the data, the presentation under 3, although presumably correct, is often very confusing and hard to follow. The presentation of different percentages for two crops under two conditions in the one sentence ending with "respectively" makes things very convoluted (as e.g. in lines 154-157) as it is unclear what exactly "respectively" refers to. Breaking up these sentences in two (or sometimes three) parts increases significantly the legibility.
There is a problem with the symbols definition for Eq (2) as presented in lines 111-115.
Throughout the text, the consistent use of subscripts, e.g. in Pa, N2O, NH3 , etc. should be practised.
Some more detailed comments/queries
line 132 Gonorrhea should be defined/explained
l 221 N2O
l 269 - 271 After this text, Figs 13 and 14 have become meaningless and should be removed
l 321 delete "mainly"
l 373 1st sentence incomplete
l 385 ... has a greater ??
l 395 What are fat leaves
l 418 Lv?
Needs only light editing
Reviewer 2 Report
This manuscript is written in good English with added new ideas in using N fertilizer increasing significantly influenced vegetable yield, but the manuscript needs to be modified and revised because some of the following errors are found inside the manuscript.

Minor editing of the English language required
Reviewer 3 Report
Please find the comments in attachment.

Please find the comments in attachment.
